

**Semantics about soil organic carbon storage: DATA4C+, a comprehensive**
**thesaurus and classification of management practices in agriculture and**
**forestry**
Kenji FUJISAKI[1], Tiphaine CHEVALLIER[2], Antonio BISPO[1], Jean-Baptiste
LAURENT[3,4], François THEVENIN[5], Lydie CHAPUIS-LARDY[2,6], Rémi
CARDINAEL[3,7,8], Christine LE BAS[1], Vincent FREYCON[9,10], Fabrice BENEDET[9,10],
Vincent BLANFORT[11,12], Michel BROSSARD[2], Marie TELLA[13,14], Julien
DEMENOIS[3,15,16]*
INRAE, InfoSol, 45075, Orléans, France
UMR Eco&Sols, IRD, INRAE, CIRAD, Institut Agro, Univ Montpellier, Montpellier,
France
AIDA, Univ Montpellier, CIRAD, Montpellier, France
CIRAD, UPR AIDA, 34398 Montpellier Cedex 5, France
Société Khaméos
LMI IESOL, Dakar, Sénégal
CIRAD, UPR AIDA, Harare, Zimbabwe
Department of Plant Production Sciences and Technologies, University of
Zimbabwe, Harare, Zimbabwe
Forêts & Sociétés, Univ Montpellier, CIRAD, Montpellier, France
CIRAD, UPR Forêts & Sociétés, F-34398 Montpellier, France
SELMET
CIRAD, SELMET
US Analyses, Univ Montpellier, CIRAD, Montpellier, France
CIRAD, US Analyses, 34398 Montpellier Cedex 5, France
CIRAD, UPR AIDA, Turrialba 30501, Costa Rica
CATIE, Centro Agronómico Tropical de Investigación y Enseñanza, Turrialba
30501, Costa Rica
*correspondence to : Julien Demenois (julien.demenois@cirad.fr)



**Abstract**
Identifying the drivers of soil organic carbon (SOC) stock changes is of utmost
importance to contribute to global challenges like climate change, land degradation,
biodiversity loss or food security. Evaluating the impacts of land-use and management
practices in agriculture and forestry on SOC is still challenging. Merging datasets or
making databases interoperable is a promising way but still with several semantic
challenges. So far, a comprehensive thesaurus and classification of management
practices in agriculture and forestry is lacking, especially while focussing on SOC
storage. Therefore, the aim of this paper is to present a first comprehensive thesaurus
for management practices driving SOC storage (DATA4C+). The DATA4C+ thesaurus
contains 226 classified and defined terms related to land management practices in
agriculture and forestry. It is organized as a hierarchical tree reflecting the drivers of
SOC storage. It is oriented to be used by scientists in agronomy, forestry and soil
sciences with the aim of uniformizing the description of practices influencing SOC in
their    original    research.    It    is    accessible    in    Agroportal
(http://agroportal.lirmm.fr/ontologies/DATA4CPLUS)  to  enhance  its  findability,
accessibility, interoperability and re-use by scientists and others such as laboratories
or land managers. Future uses of the DATA4C+ thesaurus will be crucial to improve
and enrich it, but also to raise the quality of meta-analyses on SOC, and ultimately help
policy-makers to identify efficient agricultural and forest management practices to
enhance SOC storage.

**Keywords**
interoperability, data, FAIR movement, climate change, soil carbon sequestration



## 1. Introduction

Soil organic carbon (SOC) represents about 25% of the potential of natural climate solutions (NCS) to mitigate climate change (Bossio et al., 2020). Maintaining or increasing SOC stocks can play a significant role to tackle global challenges like climate change, but also land degradation, biodiversity loss or food security (IPCC, 2019). Identifying and addressing the drivers of SOC stock changes is therefore crucial to contribute to Sustainable Development Goals (e.g. SDGs 2, 13 and 15) adopted by the United Nations in 2015 (UN General Assembly, 2015).

Wiesmeier et al. (2019) reported a large number of drivers at various scales, from climate to soil physico-chemistry, including land-use and management practices. Land-use and management practices shape carbon inputs and outputs at the plot scale, quality of carbon inputs, and may modify the turnover of soil organic matter (SOM) and SOC stocks (e.g. Fujisaki et al., 2018; Paustian et al., 2016; Poeplau et Don, 2015; Powlson et al., 2016). Evaluating the efficiency of management practices (e.g. no tillage, organic amendments) and improving our understanding of processes involved in SOC storage is still challenging and discussed (Chenu et al., 2019; Erb et al., 2017). Consequently, large datasets are necessary to make statistically robust analysis of SOC storage and its drivers. In that perspective, the number of systematic reviews or meta-analyses is growing (e.g. Beillouin et al., 2021; Bolinder et al. 2020; Cardinael et al., 2018; Fujisaki et al. 2018). Data-driven soil research and the inference of soil knowledge directly from data by using computational tools and modelling techniques, are becoming more and more popular (Wadoux et al., 2020). Merging datasets or making databases interoperable to have global datasets is another promising way (e.g. Lawrence et al., 2020; Wieder et al., 2020). Open Science (OCDE, 2015) and the FAIR



– i.e. Findability, Accessibility, Interoperability, Reusability- guiding principles
(Wilkinson et al., 2016) offer opportunities to explore this path.
However, two conditions for drivers, such as land-use and management practices, are
compulsory for systematic reviews, meta-analyses or interoperability of databases on
SOC storage: 1) have standard definitions and 2) be homogeneously described.
Harden et al. (2018) highlighted the need for harmonized description of land-use and
management practices. Todd-Brown et al. (2021) emphasized the role that semantics
should play to overcome the challenges above. Indeed, there are currently two major
limitations for these drivers of SOC change: subjectivity of the semantics and limited
scope of the terms. Many global scale studies do not always clearly define the
management practices, and use subjective terms like "improved management", or
"best management practices" (Batjes, 2019; Paustian et al., 2016; Smith et al., 2020).
Consequently, comparisons between studies might be impossible as improvement or
best management practices are highly context dependent (i.e. agronomic, climatic,
socioeconomic, or time context) (Rosenstock et al., 2016). Reversely, meta-analyses
or original studies that evaluate the effect of specific land management practices on
SOC storage provide detailed description of the land-use and management practices
but their scope is generally limited to one land cover type, one broad category of land
management practice, or focus on a climatic zone, a region or a country (Cardinael et
al., 2018; Corbeels et al., 2019; Li et al., 2018; Poeplau and Don, 2015; Maillard and
Angers, 2014).
Several standards are available for the description of land cover (e.g. FAO Land Cover
Classification System, System of Environmental-Economic Accounting (SEEA)) and
more recently of land-use (e.g. Intergovernmental Panel on Climate Change, SEEA)
(Jansen and DiGregorio, 2002; Pesce et al., 2018). Three standards for farming





practices are listed by the Agrisemantics map of data standards (Pesce et al., 2018):
a    list    of    agricultural    practices    established    by    the    FAO
(https://vest.agrisemantics.org/node/20351), the land-use categories in World Census
of Agriculture (https://vest.agrisemantics.org/node/20353), and the SEEA Land-use
Classification (https://vest.agrisemantics.org/node/20352). However, a comprehensive
thesaurus and classification of management practices is lacking, especially while
focussing on SOC storage. For instance, the standards for " farming practices " listed
in the Agrisemantics map (https://vest.agrisemantics.org/by-theme/7705/7705/7713)
are not exhaustive (e.g. empirical farmers' practices in Southern countries), nor
harmonized or/and specific to SOC storage. As far as we know, there has been no
attempt to deal with these shortcomings to be able to understand, quantify or
extrapolate processes and drivers of SOC storage in agriculture and forestry using
large databases. Therefore, the objectives of this study were: i) to compile a
comprehensive thesaurus, i.e. a list of standards and specifically defined terms, for
management practices driving SOC storage, ii) to keep such thesaurus easy to use for
non-scientists such as soil test laboratories or land managers, and iii) to define a
classification of these drivers to further enhance interoperability of databases on SOC.
The aim of this paper is to present a first comprehensive thesaurus and classification
of management practices in agriculture and forestry with a focus on soil organic carbon
called DATA4C+.





## 2. Materials and Methods

### 2.1. Identification of SOC drivers related to land management practices

In the present work, land management practices covered croplands, grasslands and forestry practices established at the field scale, without any change in land-use. We identified land management practices which are recognized in scientific literature to influence SOC change. Original papers (e.g. Cardinael et al., 2018; Mayer et al., 2020; Smith et al., 2020, see Table 1 for some examples and Supplementary material for the full list), technical and institutional reports (e.g. Chotte et al., 2019; Pellerin et al., 2020; Sanz et al., 2017; Smith et al., 2007) were used to identify these land management practices.

Only land management practices explicitly described were retained. Therefore, management practices labelled as "improved" were discarded. Consequently, we included practices considered as nominal or conventional (e.g. monoculture, conventional tillage). Agroforestry was considered in this study as a land management practice, since it is defined as an agroecosystem where "forest species of trees and other wooded plants are purposely grown on the same land as agricultural crops or livestock, either concurrently or in rotation" (FAO, 2015).

### 2.2. Definition of drivers

Definitions of land cover classes, land-use classes, and land management practices were found in data standards (e.g. World Census of Agriculture, FAO, 2015), thesaurus



and scientific literature collected at the former step of driver identification. In case a
definition was lacking in the primary data source, it was collected through thematic
glossaries (e.g. IPCC, 2019; "Landmark Glossary"; "WOCAT Glossary").

## 2.3. Classification of land management practices

As there is currently no comprehensive thesaurus for land management practices
which directly or indirectly affect SOC dynamics, we classified the single management
practices gathered in the previous steps into a hierarchical tree. This hierarchical tree
was built thanks to existing classifications of land management practices found in
literature. These classifications usually rely on the manipulation of several components
of the agroecosystem which often affect C inputs and C outputs from soils, such as the
plant management, water management or soil tillage management for example (Table
1). We considered, in the hierarchical tree, only single land management practices.
Integrated land management practices (e.g. conservation agriculture, organic
agriculture) were not included as a whole, but described by their single components
(e.g. conservation agriculture = no tillage, permanent soil cover, rotation/crop
diversification).

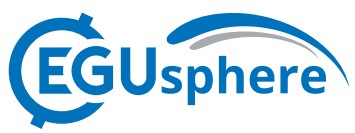



Table 1. Examples of land-use change and land management practices classification for the assessment of soil organic carbon stock change (see supplementary material for the full list used)

| Reference | Forests | Annual and perennial croplands | Grasslands | Land-use change |
|---|---|---|---|---|
| (Smith et al., 2008) IPCC report for GHG mitigation in agriculture | | - Improved agronomic practices<br>- Nutrient management<br>- No till & residue retention<br>- Water management<br>- Manure application | | |
| (Paustian et al., 2016) Land management practices for climate-smart soils | | - Add nutrients; add lime; grow N fixing species<br>- Grow cover crops; reduce or vegetate fallow fields<br>- Reduce to economic-optimal rates<br>- Reduce or halt tilling; implement residue retention<br>- Improve timing and placement; use enhanced efficiency fertilizer<br>- Rotate perennials; use agroforestry; use high-C input species; grow cover crops<br>- Add amendments such as compost and biochar | - Convert to perennial vegetation<br>- Restore to wetland | |
| (Griscom et al., 2017) Evaluation of land management practices for GHG mitigation | - Natural forest management<br>- Improved plantations<br>- Avoided woodfuel<br>- Fire management | - Biochar<br>- Trees in croplands<br>- Nutrient management<br>- Conservation agriculture<br>- Improved rice | - Grazing-feed<br>- Grazing-animal management<br>- Optimal intensity<br>- Legumes | - Reforestation<br>- Avoided forest conversion<br>- Avoided grassland conversion |





| (Chotte et al., 2019) Sustainable land management practices for land degradation neutrality | - Agroforestry<br>- No/minimum tillage<br>- Crop rotation<br>- Intercropping<br>- Green manuring<br>- Composting/mulching<br>- Manuring<br>- Integrated crop/livestock systems<br>- Conservation agriculture<br>- Fertilizer use | - Reduce herd densities | - Afforestation<br>- Reforestation |



| (Smith et al., 2020) Land management practices for food security, climate change mitigation, and against desertification and land degradation | Improved forest management refers to management practices in forests for the purpose of climate change mitigation. It includes a wide variety of practices affecting the growth of trees and the biomass removed, including improved regeneration (natural or artificial) and a better schedule, intensity, and execution of operations (thinning, selective logging, final cut; reduced impact logging, etc.). | - Improved cropland management is a collection of practices consisting of (a) management of the crop: including high carbon input practices, for example, improved crop varieties, crop rotation, use of cover crops, perennial cropping systems, integrated production systems, crop diversification, agricultural biotechnology; (b) nutrient management: including optimized fertilizer application rate, fertilizer type (organic manures, compost, and mineral), timing, precision application, nitrification inhibitors; (c) reduced tillage intensity and residue retention; (d) improved water management: including drainage of waterlogged mineral soils and irrigation of crops in arid/ semiarid conditions; (e) improved rice management: including water management such as mid-season drainage and improved fertilization and residue management in paddy rice systems; and (f) biochar application <br><br> - Practices that increase soil organic matter content include a) land-use change to an ecosystem with higher equilibrium soil carbon levels ; (b) management of the vegetation: including high carbon input practices, for example, improved varieties, rotations and cover crops, perennial cropping systems, biotechnology to increase inputs and recalcitrance of below ground carbon; (c) nutrient management and organic material input to increase carbon returns to the soil: including optimized fertilizer and organic material application rate, type, timing, and precision application; (d) reduced tillage intensity and residue retention; and (e) improved water management: including irrigation in arid/semiarid conditions | Improved grazing land management is a collection of practices consisting of (a) management of vegetation: including improved grass varieties/sward composition, deep rooting grasses, increased productivity, and nutrient management; (b) animal management: including appropriate stocking densities fit to carrying capacity, fodder banks, and fodder diversification; and (c) fire management: improved use of fire for sustainable grassland management, including fire prevention and improved prescribed burning (see also fire management as a separate practice below) | - Reduced grassland conversion to cropland <br> - Reduced deforestation and degradation <br> - Reforestation and forest restoration <br> - Afforestation <br> - Land-use change to an ecosystem with higher equilibrium soil carbon levels (e.g., from cropland to forest) |
|---|---|---|---|---|



| | | |
|---|---|---|
| (Bai et al., 2019) Effect of climate-smart agriculture practices on soil carbon stocks | - Conservation tillage<br>   - no-till<br>      - reduced tillage<br>- Cover crops<br>- Biochar<br>- Other agronomic practices: crop residues, nitrogen fertilization, irrigation, and crop rotation | |
| (Chambers et al., 2016) 4P1000 potential in the USA | - Conservation cover<br>- Conservation crop rotation<br>- Residue and tillage management, no-till<br>- Strip till<br>- Contour farming<br>- Contour buffer strips<br>- Residue and tillage management, reduced till<br>- Field border<br>- Filter strips<br>- Grassed waterways<br>- Strip-cropping<br>- Vegetative barriers<br>- Herbaceous wind barriers | - Forage and biomass planting<br>- Prescribed grazing<br>- Range planting |
| (Corbeels et al., 2019) 4P1000 potential in sub-Saharan Africa through agroforestry and conservation agriculture | Conservation agriculture:<br>- Minimum/no tillage<br>- Minimum/no tillage + residues<br>- Minimum/no tillage + residues + intercropping or rotation<br><br>Agroforestry:<br>Alley cropping<br>Multistrata systems<br>Fallows<br>Parklands | Parklands |





| | | | |
|---|---|---|---|
| (Pellerin et al., 2019) 4P1000 potential in mainland France | - No-tillage<br>- Cover crops<br>- Increase of temporary grasslands in crop rotations<br>- Increase exogenous organic matter application<br>- Agroforestry<br>- Hedgerows<br>- Cover crops in vineyards | - Moderate intensification of grasslands: fertilization, increase leguminous species, increase grass export<br>- Haying rather than grazing | - Cultivation to grass<br>- Native to grass |
| (Conant et al., 2017) Effect of grassland management on soil carbon stocks | | - Fertilization<br>- Fire<br>- Grazing<br>- Grass ley<br>- Reclamation | |
| (Batjes, 2019) Effect of grassland management on soil carbon stocks | | - Controlled grazing<br>- Adjusting stocking rates<br>- Improved pastures with leguminous crops<br>- Fire management | |
| (Mayer et al., 2020) Effect of forest management on soil carbon stocks | - Nitrogen addition<br>- Selection of species with N-fixing associates<br>- Trees species selection<br>- Management of tree species diversity<br>- Management of stand density and thinning<br>- Removal of forest residues<br>- Herbivory regulation<br>- Fire management | | Afforestation |






| (Cardinael et al., 2018) IPCC Tier 1 coefficients for agroforestry systems | - Alley cropping<br>- Fallows<br>- Hedgerows<br>- Multistrata systems<br>- Shaded perennial-crop systems<br>- Silvo-arable systems<br>- Parklands | - Parklands<br>- Silvopastures<br>- Hedgerows |
|---|---|---|



## 2.4. Design and quality control of the thesaurus

From October 2019 to October 2020, participants to the project DATA4C+ (https://www.data4c-plus-project.fr/en) carried out the editing phase of the thesaurus. Participants were junior and senior scientists from 3 French research institutions (i.e. Cirad, INRAE, IRD) that joined their expertise about organic carbon dynamics in temperate and tropical soils. A first version of the thesaurus and classification was shared and discussed among them in October 2020. The consolidation phase was carried out from November 2020 to June 2021. A second version of the thesaurus and classification was shared, discussed and validated among participants of the project in July 2021. From July 2021 to September 2021, editors of the thesaurus checked its consistency before its first available on-line version, as presented in this paper (see Fig. 1).

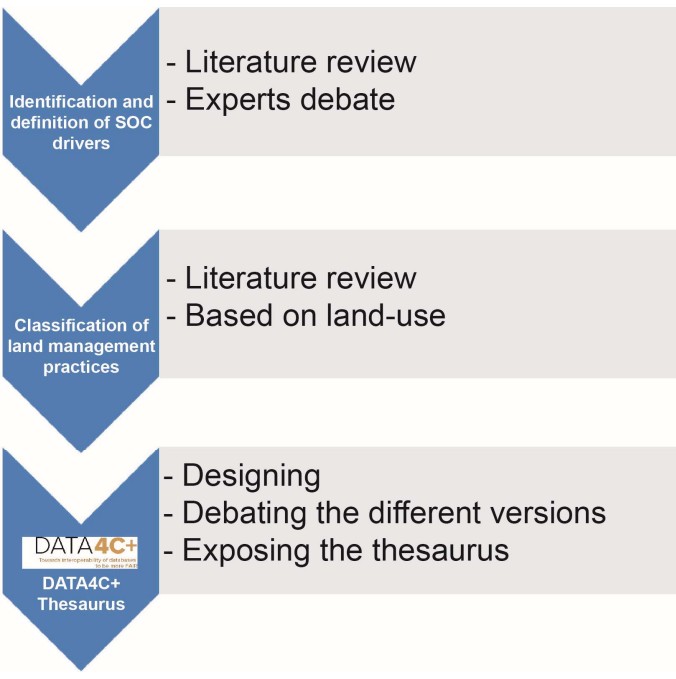





*Figure 1 – Summary of the different steps to build the DATA4C+ thesaurus*

## 3. Results

### 3.1. Land management practices

Land management practices were classified in three main categories according to
land-use: i) land management practices in annual and perennial croplands, ii) land
management practices in grasslands, and iii) land management practices in forests
and tree plantations. We chose to classify the land management practices inside large
categories of land-use rather than land cover for several reasons. Land-use categories
are well harmonized between different standards (FAO, IPCC, SEEA, World Census
of Agriculture, see Gong et al., 2009), whereas the matching of land cover categories
between the main standards is less straightforward (see, for instance, Herold et al.
(2009) and Yang et al. (2017) for the harmonization of FAO Land Cover Classification
System with other land cover standards). Land-use categories suit well with
greenhouse gas (GHG) balance accounting thanks to the IPCC framework (IPCC,
2006). Furthermore, some management practices may induce a change in land cover
without changing in land-use, such as management practices regarding plant
management like agroforestry practices.
In these categories, several sub-categories were created, regarding plant, biomass
(through grazing and animal management in grassland, residue management in
croplands, biomass fluxes in forests), and amendments management, but also erosion,
water, fire, and land clearing management in the case of agroecosystems implanted
after land clearing. These sub-categories are mainly inspired from Smith et al. (2020).
They rely on management techniques from the point of view of the land managers,



which is commonly used in literature for the classification of land management
practices that affect SOC dynamics (Table 1). Another classification of land
management practices could be specifically based on the mechanisms affecting SOC
dynamics, i.e. modification of carbon inputs and/or modification of SOM turnover.
However, this approach would be less handy for a non-scientific audience.
Furthermore, there are still knowledge gaps regarding the processes involved in SOC
sequestration after the establishment of several management practices (Chenu et al.,

216   2019).


## 218   3.2. The DATA4C+ thesaurus: technology, content and browsing

The DATA4C+ thesaurus is freely available at the following URL address:
http://data4c-plus.net/admin/thesaurus/index.
The DATA4C+ thesaurus is connected to a PostgreSQL® database. The intuitive web
interface uses the jsPlumbTree function of the jQuery library, which is a plugin that
renders a reducible and extensible tree structure representing the hierarchical
relationship between different nodes. In addition, the plugin uses the jsPlumb library to
draw connection lines using Bézier curves between nodes. The tree is drawn
dynamically from left to right and top to bottom when connecting to the database.
Each term of the database is defined by four nodes:
• data-id: term identifier. Must be unique throughout the tree
• data-parent: identifier of the parent node
• data-first-child: identifier of the first child node
• data-next-sibling: identifier of the next sibling node



The DATA4C+ thesaurus was developed by Cirad. All the source programs are
available on the forge https://gitlab.com/ecosols and can be freely accessed on request
under the CC BY-SA 4.0 FR license. To facilitate re-use of the DATA4C+ thesaurus, it
can be downloaded as Simple Knowledge Organisation System (SKOS) format (W3C,
2009).    The    DATA4C+    thesaurus    is    accessible    in    Agroportal
(http://agroportal.lirmm.fr/ontologies/DATA4CPLUS)    to    enhance    its    findability,
accessibility, interoperability and reusability by scientists in agronomy, forestry and soil
sciences. It may also be used by other end-users such as soil test laboratories to
describe the soil samples analysed or by land managers to describe and report their
practices (e.g. for carbon farming programmes). Additionally, the Comma Separated
Values (CSV) file of DATA4C+ thesaurus is available on the data depository of Cirad
(https://dataverse.cirad.fr)    under    the    CC-BY    4.0    FR    license    with    the    DOI:
https://doi.org/10.18167/DVN1/HMCPMF. The DATA4C+ thesaurus classifies 226
defined terms related to land management practices in agriculture and forestry. It is
organized as a hierarchical tree reflecting the drivers of SOC storage. To have access
to the definition of a given term, the user must find the term in the tree and click on it.
Then a "pop up" appears with the definition of the term and the source of the definition
(Fig. 2). A link to the source of the definition (URL or DOI) is given for each term. By
clicking on this link, a new web page appears.





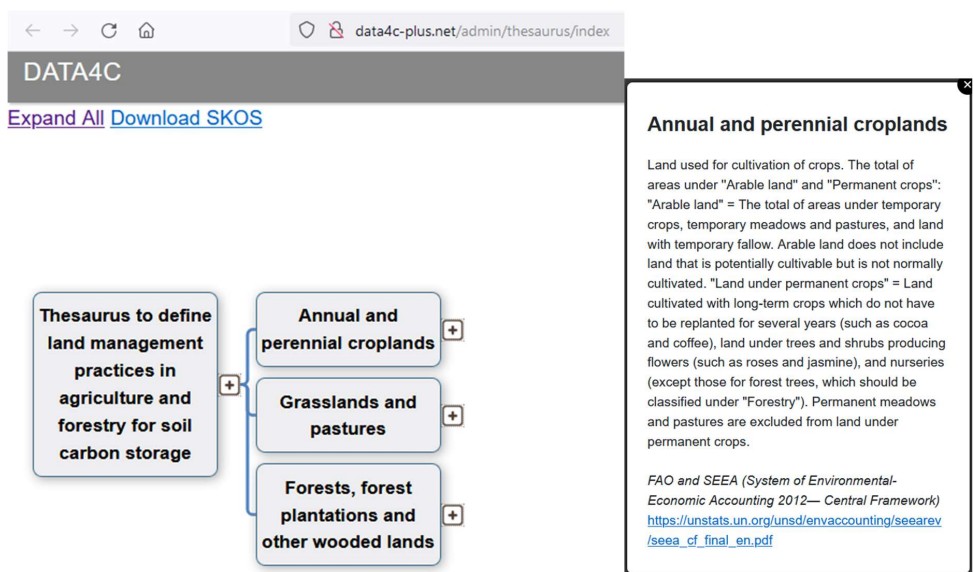


**Figure 2 – Browsing hierarchical tree and definition in the DATA4C+ thesaurus**


(http://data4c-plus.net/admin/thesaurus/index)


## 4. Discussion


### 4.1. Less subjectivity of land-use and management practices will


### improve re-use of data and quality of meta-analyses


The terms "improved management practice" or "conventional agricultural" are currently


used in the scientific literature despite their subjectivity (Sumberg & Giller, 2022). The


use of this term implicitly means comparing one practice to another practice and


describing the improved actions, which is hardly ever done. The DATA4C+ thesaurus


gives a framework to describe the practices. This is vital to produce robust meta-


analyses. For instance, the term "improved management of pastures" encompasses


diverse agronomic practices (e.g. introduction of leguminous species, switch from


mineral to organic fertilizers, no burning for land clearing, reduced grazing intensity).


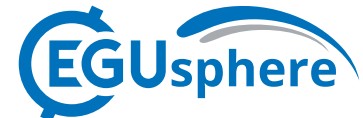

The description of each of these agronomic practices is specific: species' names and
plant density for the introduction of leguminous, type, amount and date of application
of fertilizers for the switch from mineral to organic fertilizers, amount of biomass left on
site for no burning for land clearing. Besides, their impacts on SOC stocks are highly
different as highlighted by Maia et al. (2009), Conant et al. (2017), or Fujisaki et al.

270   (2018).


**272   4.2. More genericity in the description of management practices**

**273        will improve re-use of data and quality of meta-analyses**

The DATA4C+ thesaurus intends to facilitate data sharing for the evaluation of soil
carbon storage through land management practices, thanks to the genericity of the
proposed terms. We evaluate the DATA4C+ thesaurus against land management
practices used in several meta-analyses (Table 2). In many situations, there is an
adequate matching between terms used in the meta-analyses and terms used in the
thesaurus.
However, some studies use levels of details uncovered in the thesaurus, such as the
species family of plants sown in the fields (Bai et al., 2019), or several tillage
techniques (Jian et al., 2020), that can be grouped into larger categories used in the
thesaurus (conventional vs reduced tillage). These very detailed levels were not
covered in the thesaurus because of the current lack of the evaluation of their effect
on SOC dynamics. Indeed, the effect of soil tillage on soil carbon storage is still
discussed by soil scientists (Chenu et al., 2019), and the use of numerous categories
of tillage practices may weaken the significance of the observed trends.
On the other hand, several studies use broader categories than in the present
thesaurus, which may prevent re-use of the dataset. This is the case for land



management practices in grasslands studied by Conant et al. (2017), where categories
such as "grazing" and "fire" are not further detailed, despite the wide response range
of soil carbon stocks according to the intensity of grazing for instance (Abdalla et al.,

293    2018).

Concerning meta-analyses of SOC, Beillouin et al. (2022) identified issues of low
transparency, reproducibility, and updatability. Improving the quality and reliability of
synthesis papers is of utmost importance as they are increasingly used to inform policy
decisions with possibly large environmental and socioeconomic implications (Krupnik
et al., 2019). Nosek et al. (2015) noted that advances must be made to give full and
unbiased access to scientific data in line with open science practices. In that
perspective, the transparency and the genericity of the terms defined in the DATA4C+
thesaurus, mostly inventoried in original papers, technical and institutional reports, will
contribute to increase the quality of data and ultimately to merge and analyze data from
various sources.





Table 2. Matching evaluation of land management practices assessed in meta-analyses against land management practices in the
DATA4C+ thesaurus.

| Source | Land management category in paper | Land management practice evaluated | Land management practice or variable in the DATA4C+ thesaurus |
|---|---|---|---|
| Bai et al. (2019) | Climate Smart Agriculture practices | No-till | No-till |
| | | Reduced tillage | Reduced tillage or minimum tillage |
| | | Cover crop | Cover crop |
| | | Biochar | Biochar |
| | Crop residue | Return | Mulched residues OR Shredded residues OR Buried residues |
| | | Remove | Exported residues |
| | Nitrogen fertilization | 1-100 | Partially covered: mineral fertilization practice is included but not the quantity supplied |
| | | 101-200 | Partially covered: mineral fertilization practice is included but not the quantity supplied |
| | | > 200 | Partially covered: mineral fertilization practice is included but not the quantity supplied |
| | Water management | Irrigation | Irrigation |
| | Crop sequence | Rotational | Rotation of annual crops |
| | | Continuous | Monoculture |
| | Cover crop species | *Poaceae* | Not covered in the thesaurus |




| Reference | Category | Practice | Comment |
|---|---|---|---|
| Conant et al. (2017) | Grassland management | *Fabaceae* | Not covered in the thesaurus |
| | | *Poaceae + Fabaceae* | Not covered in the thesaurus |
| | | Fertilizer | Mineral fertilization |
| | | Grazing | Several choices required in "Grazing management" |
| | | Sowing improved grass species | Plant breeding |
| | | Grass ley in rotation | Temporary grassland in crop rotation |
| | | Fire | Several choices in required in "Fire management" |
| | | Earthworms | Not covered in the thesaurus |
| | | Irrigation | Irrigation |
| | | Reclamation | Not covered in the thesaurus |
| | | Silvopastoralism | Silvopastures |
| Shi et al. (2018) | Agroforestry practices | Alley cropping | Alley cropping |
| | | Homegardens | Multistrata systems |
| | | Silvopastures | Silvopastures |
| | | Windbreaks | Hedgerows |
| Han et al. (2016) | Crop fertilization | Unbalanced application of chemical fertilizers | Partially covered: mineral fertilization practice is included but not the appreciation of balanced vs unbalanced application |
| | | Balanced chemical fertilization | Partially covered: mineral fertilization practice is included but not the appreciation of balanced vs unbalanced application |
| | | Straw retention and | Mulched residues OR Shredded residues OR Buried residues AND |



| | | | application of chemical fertilizers | Mineral fertilization |
|---|---|---|---|---|
| | | | Application of manure and chemical fertilizers | Solid manure OR liquid manure AND Mineral fertilization |
| Jian et al. (2020) | Tillage group | | Disk tillage | Conventional tillage |
| | | | Sweep | Conventional tillage |
| | | | Tandem disk | Conventional tillage |
| | | | Full-tilled | Conventional tillage |
| | | | Mouldboard ploughing | Conventional tillage |
| | | | Harrowing | Conventional tillage |
| | | | Moldboard plowing | Conventional tillage |
| | | | Turnplow | Conventional tillage |
| | | | Plow-till | Conventional tillage |
| | | | Ridge-till | Ridge tillage |
| | | | Mulch tillage | Reduced tillage or minimum tillage |
| | | | Chisel | Reduced tillage or minimum tillage |
| | | | Slit tillage | Reduced tillage or minimum tillage |
| | | | Light tillage | Reduced tillage or minimum tillage |
| | | | Strip-tiller tillage | Strip tillage |
| | | | Deep-till | Conventional tillage |
| | | | No-tillage | No-till |



| Jian et al. 2020 | Conservation type | Agriculture forest system | Several choices required in the category Agroforestry |
| --- | --- | --- | --- |
| | | Cover crop | Cover crop |
| | | No tillage | No-till |
| | | Reduced tillage | Reduced tillage or minimum tillage |
| | | Organic farm | Organic agriculture |
| | | Straw return, mulching | Mulched residues |
| | | Stubble | Not covered in the thesaurus |
| | | Ridging | Ridge tillage |
| | | Rotation | Rotation of annual crops |
| | | Plastic film mulching | Not covered in the thesaurus |
| | | Interplanting | Intercropping |
| | | Combination of two | Not covered in the thesaurus |
| | | Organic farm with cover crop as green manure | Organic agriculture AND Cover crop |
| | | Organic farm with no tillage | Organic agriculture AND No-till |





### 4.3. Future development of the DATA4C+ thesaurus: uses and accrual


The DATA4C+ thesaurus is expected to be used by scientists in agronomy, forestry
and soil sciences with the aim of uniformizing the description of practices influencing
SOC in their original research. As it was developed to be simple and easy-to-use, the
thesaurus may also be used by several end-users as land managers (e.g. to report
their practices for carbon farming) or by laboratories to describe the soil samples
analysed (e.g. metadata on the sample). The generated data will therefore be more
easily to retrieve and to be integrated to perform meta-analyses in particular. Another
perspective will be to mobilize the DATA4C+ thesaurus to feed models on SOC
dynamics with more site-specific data. However, such perspective would need to
enrich the DATA4C+ thesaurus with vocabulary related to annual carbon inputs to
enhance carbon inputs to soil (e.g. Bolinder et al., 2007). Accrual of the DATA4C+
thesaurus could also be focused on emerging practices and empirical farmers'
practices, which are poorly studied by researchers. Peer-reviewing of the updated
versions of the DATA4C+ thesaurus will be performed by the Scientific and Technical
Committee of the 4 per 1000 Initiative (https://4p1000.org/). Versioning of the
DATA4C+ thesaurus will be done at the following URL address: http://data4c-
plus.net/admin/thesaurus/index,                    in                    Agroportal
(http://agroportal.lirmm.fr/ontologies/DATA4CPLUS) and on the data repository of
Cirad (https://doi.org/10.18167/DVN1/HMCPMF). Suggestions of accrual could be
sent to the corresponding author or at the following email address: data4c@cirad.fr .

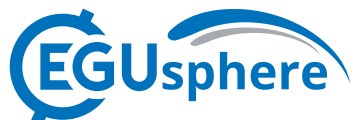

**5. Conclusion**
The DATA4C+ thesaurus is the first attempt to compile and classify the land-use and
management practices in agriculture and forestry influencing SOC storage. Future
uses of the DATA4C+ thesaurus will be crucial to improve and enrich it, but also to
raise the quality of meta-analyses on SOC, and ultimately help policy-makers to identify
efficient agricultural and forest management practices to improve SOC storage. In that
sense, the DATA4C+ thesaurus is a contribution to SDG 17 "Partnerships for the goals"
(i.e. goals 17.6 and 17.7).

**Appendix**
Supplementary material: the full list of references, technical and institutional reports
used to identify the land management practices.

**Code availability**
The DATA4C+ thesaurus was developed by Cirad and Khaméos. All the source
programs are available on the forge https://gitlab.com/ecosols and can be freely
accessed on request under the CC BY-SA 4.0 FR license.

**Data availability**
The        DATA4C+        thesaurus        is        accessible        in        Agroportal
(http://agroportal.lirmm.fr/ontologies/DATA4CPLUS).    The    CSV    file    of    DATA4C+
thesaurus    is    available    on    the    repository    of    Cirad    in    the    Dataverse    CIRAD



(https://dataverse.cirad.fr) under the CC-BY 4.0 FR license with the DOI:
https://doi.org/10.18167/DVN1/HMCPMF.

**Author contributions**
Kenji Fujisaki led the inventory and analyses of resources to build the thesaurus.
François Thévenin (Khaméos) and Jean-Baptiste Laurent did the informatic
development of the thesaurus. Antonio Bispo, Tiphaine Chevallier and Julien
Demenois supervised the conceptualization of the thesaurus. Kenji Fujisaki and Julien
Demenois prepared the manuscript with contributions from all the co-authors. All the
co-authors reviewed the thesaurus and the manuscript.

**Competing interest**
The authors declare that they have no conflict of interest.

**Acknowledgements**
We would like to thank colleagues from Agroportal for their technical help in publishing
the DATA4C+ thesaurus.

**Financial support**
Support was provided by the French National Research Agency (ANR, https://anr.fr/)
through the project DATA4C+ "Towards interoperability of databases to be more FAIR"
(https://www.data4c-plus-project.fr/en) (PROJECT N° ANR-19-DATA-0005-01). The
DATA4C+ project is led by the CIRAD-INRAE-IRD consortium. The funders had no



role in study design, data collection and analysis, decision to publish, or preparation of
the manuscript.

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
