# Peer review of "Semantics about soil organic carbon storage: DATA4C+, a comprehensive"

_EGUsphere, 2022_

## Author Comment (AC1)

This paper describes a first attempt to synthesise definitions of interventions known from the literature to affect soil organic carbon. The output of this is an openly available database, DATA4C+, which can be accessed online and downloaded. The target end users include researchers and land managers. The objective is to improve consistency in the terms used to describe land management interventions to assist the ease of use of experimental data in e.g. metanalyses.

The aim is commendable and consistency in definitions and description will undoubtedly aid more powerful analyses of data to be conducted. Since this is a descriptive manuscript rather than experimental, this is reflected in the review.

This work is of scientific value, and as the first attempt of its kind represents a novel contribution. It is well presented and the language is very well written.

It certainly falls within the scope of the journal and should be of broad international interest and therefore warrants consideration for publication.

There are a few points I would like to address to the authors:

1. How was the initial literature search conducted? There is information regarding how certain definitions were excluded, but not how papers were searched in the first instance.

Thank you for your comment. The literature search was conducted based on expert knowledge and not systematic review. Therefore a first list of meta-analyses was established by the authors. This list is available in Table 1. Focus was put on meta-analyses as homogeneous definitions are a pre-requisite to conduct such analyses. Besides, the list of definitions gathered from the meta-analyses was completed by definitions coming from glossaries (e.g. IPCC) and existing thesaurus (e.g. Agrovoc), which are hardly referenced in search engines like Scopus, Web of Science or Google Scholar. Finally, this list of definitions was extensively discussed among the group of authors resulting in the selection of other definitions than the initial ones and inclusion of new definitions.

2. I am not sure what value Table 1 gives? I don't think I understand it (my fault, probably) but think it needs explaining in more detail or excluding.

We do agree with your comment.Therefore, the current Table 1 will be inserted as supplementary material in the revised version of the manuscript.

3. Was there a quantitative method for 'testing' the thesaurus? It seems that there were discussions around it but perhaps where was a more structured way of testing the definitions. Can this be explained further?

Thank you for your question. In this work we did not quantitatively test the thesaurus. We would be more than happy to discuss this point with you to find a way to do so ! At that stage, our approach is more qualitative and iterative. Qualitative because if the thesaurus is used in future studies, than we could see this as a positive test : the thesaurus and its definitions meet the users' needs. Iterative because we are aware that this first version of the thesaurus is not perfect and some definitions are probably not fully consensual. Therefore, the amount of comments and suggestions of new terms and definitions will be another indicator to test the thesaurus.

4. The gatekeepers are listed as Scientific and Technical Committee of the 4 per 1000 Initiative. How often will these meet to discuss new entries? Will previous entries be reviewed?

Usually, the Scientific and Technical Committee (STC) of the 4 per 1000 Initiative gathers 2 or 3 times a year. It seems reasonable to assume that the thesaurus could be put at the agenda once a year. The previous entries were not formally reviewed by the STC even though the thesaurus was shared with some of their members.

---

## Author Comment (AC2)

I laud the authors for undertaking the development of a thesaurus for land use/management terms as they relate to soil organic carbon. I absolutely agree that imprecise terminology hinders our field. Overall, this manuscript is in good shape after the authors have addressed the comments from the first round of review.

Thank you for your comment which reinforces our convinction that semantics is an overlooked issue.

I do disagree with the authors that all conventional tillage can be lumped together but I also understand their reasoning. There are well known differences between moldboard and shallow disk tillage (disks can cut anywhere from 5 to 20+ cm into the earth) with the former truly inverting the soil and the later only breaking up the surface.

Thank you for pointing this out and agree with your suggestion of introducing additional criteria related to the type of disturbances and intensity. Additionnaly, as terms like « conventional tillage » and « reduced tillage » are subjective, we will revised the entry « Tillage management » in the category « Annual and perennial crops » as follow :

[Figure]

With the following definitions :

- **No till** (no change in the definition) : « Zero tillage or no tillage does not involve any tillage operations on arable land. After the seeding operation, not more than 25 percent of the soil surface is allowed to be disturbed. The soils are always covered, including for the period between harvest and sowing. The stubble is retained and the soil surface is covered by residue mulch or stubble for erosion control » FAO, 2015. World Programme for the Census of Agriculture 2020. Volume 1 Programme, Concepts and Definitions. FAO, Rome, http://www.fao.org/3/i4913e/i4913e.pdf
- **Intermediate intensity tillage** : « Any non-inversion tillage practice performed above 40 cm depth (e.g. chisel tillage, disk tillage, mulch tillage). These tillage practices are often referred as "Reduced tillage", "Minimum tillage", or "Conservation tillage" » Authors DATA4C+, modified after Haddaway et al. 2017, Haddaway, N.R., Hedlund, K., Jackson, L.E., Kätterer, T., Lugato, E., Thomsen, I.K., Jørgensen, H.B., Isberg, P.-E., 2017. How does tillage intensity affect soil organic carbon? A systematic review. Environmental Evidence 6, 30. https://doi.org/10.1186/s13750-017-0108-9
- **High intensity tillage** : « Any inversion tillage practice (e.g. mouldboard ploughing), or any non-inversion tillage practice performed to 40 cm depth or below (e.g. subsoiling, very deep chisel or disk tillage). These tillage practices are often referred as "Conventionnal tillage" in the literature, and may be followed by a secondary tillage practice to prepare seedbed », Authors

DATA4C+, modified after Haddaway et al. 2017, Haddaway, N.R., Hedlund, K., Jackson, L.E., Kätterer, T., Lugato, E., Thomsen, I.K., Jørgensen, H.B., Isberg, P.-E., 2017. How does tillage intensity affect soil organic carbon? A systematic review. Environmental Evidence 6, 30. https://doi.org/10.1186/s13750-017-0108-9

- **Strip tillage** (no change in the definition) : « Strips are tilled to receive the seed while the soil along the intervening bands is not disturbed and remains covered with residues such as mulch. » FAO, 2015. World Programme for the Census of Agriculture 2020. Volume 1 Programme, Concepts and Definitions. FAO, Rome, http://www.fao.org/3/i4913e/i4913e.pdf

The term « Ridge tillage » will be deleted and could either be included in the category « Intermediate intensity tillage » or « High intensity tillage » depending on the tools and the depth of soil worked to form the ridges. Indeed, there is still some ambiguity in the scientific literature on « ridge tillage ». Haddaway et al. (2017) classified this practice as a 'high intensity tillage' practice because the soil is turned over, but this does not seem to be systematic. This practice is classified by some authors as "conventional tillage" (Jian et al. 2020), while others consider it to be "conservation tillage", e.g. Follett (2001)[1]. Angers & Eriksen-Hamel (2008)[2] also did not consider this practice to be "full-inversion tillage". We therefore consider that such classification will help avoiding confusion.

The text and the tables of the manuscript will be revised accordingly.

My only concern, and I'm not sure how this would or should even be addressed in this manuscript, is that there seems to be an assumption that just because the authors have made this thesaurus, other scientists are going to start using it. Is there a plan to promote and educate? Perhaps that is the job of the 4p1000 initiative.

Thank you for this comment. This is also definitely one of our concerns ! So far our strategy of dissemination and promotion relies on :

- The inclusion of the DATA4C+ thesaurus in AgroPortal which is a repository more visible for scientists than the current website of the thesaurus
- The dissemination and the mobilization of the thesaurus in scientific publications, and in the works of our colleagues and networks (including social media)
- The promotion by the Scientific and Technical Committee of the 4p1000 Initiative

The current Horizon Europe project ORCaSa (Operationalising the international Research Cooperation on Soil carbon) in which we are involved will be also an opportunity of dissemination. A knowledge platform will be developed and could possibily host also the DATA4C+ thesaurus.

However, all efforts and ideas are welcome !

To address this point in the manuscript, we will revised it as follow : « Promotion and peer-reviewing of the DATA4C+ thesaurus and its updated versions will be performed by the Scientific and Technical Committee of the 4 per 1000 Initiative (https://4p1000.org/). »
* * *
[1] https://www.sciencedirect.com/science/article/pii/S0167198701001805#aep-section-id15).
[2] https://acsess.onlinelibrary.wiley.com/doi/full/10.2136/sssaj2007.0342